# Association of MMP-8 -799C/T Polymorphism with Peri-Implantitis: A Cross-Sectional Study

**DOI:** 10.3390/jpm15050182

**Published:** 2025-05-01

**Authors:** Ioannis Fragkioudakis, Christine Kottaridi, Aikaterini-Elisavet Doufexi, Konstantinos Papadimitriou, Leonidas Batas, Dimitra Sakellari

**Affiliations:** 1Department of Preventive Dentistry Periodontology and Implant Biology, School of Dentistry, Aristotle University of Thessaloniki, Thessaloniki 54124, Greece; papadent3108@gmail.com (K.P.); lbatas76@gmail.com (L.B.); dimisak@gmail.com (D.S.); 2General Microbiology Laboratory, Department of Genetics, Development and Molecular Biology, School of Biology, Aristotle University of Thessaloniki, Thessaloniki 54124, Greece; ckottaridi@bio.auth.gr

**Keywords:** peri-implantitis, genetic polymorphism, MMP−8, −799C/T SNP, biomarkers, personalized dental care

## Abstract

**Purpose**: This study explored the relationship between matrix metalloproteinase−8 (MMP−8) gene polymorphisms (−799C/T, −381A/G, and +17C/G) and peri-implantitis, examining clinical parameters including the probing depth (PD), clinical attachment level (CAL), and bleeding on probing (BOP). **Methods**: This cross-sectional study involved 120 participants categorized into peri-implantitis and healthy implant groups according to the 2018 classification criteria for periodontal and peri-implant diseases. Saliva samples were analyzed for MMP−8 polymorphisms using polymerase chain reaction (PCR) and Sanger sequencing. Statistical analyses were conducted to evaluate genotype- and allele-specific risks and their associations with clinical parameters. **Results**: Among the 95 samples analyzed, the −799C/T polymorphism was significantly associated with peri-implantitis, with T allele carriers having a higher diagnosis rate (odds ratio: 3.04, *p* = 0.010). Although T allele carriers exhibited higher mean values for the probing depth (PD), clinical attachment level (CAL), and bleeding on probing (BOP), these differences were not statistically significant across genotypes. No associations were found between the −381A/G and +17C/G polymorphisms and peri-implantitis clinical parameters. **Conclusions**: The −799C/T polymorphism, specifically the T allele, is strongly linked to peri-implantitis, indicating its potential as a genetic marker for disease susceptibility. Further research is required to investigate the role of MMP-8 polymorphisms in peri-implant diseases and to advance the development of personalized diagnostic tools.

## 1. Introduction

Peri-implantitis is an inflammatory disease with multiple causes that damages the tissue around a dental implant, resulting in gradual bone loss and potentially endangering the implant’s survival [1]. Although peri-implantitis and periodontitis have similar causes (for example, colonization by Gram-negative anaerobes), the course of peri-implantitis is shaped by a multifaceted combination of environmental and genetic influences [1,2]. Among the host-related factors, polymorphisms in the matrix metalloproteinase-8 (MMP−8) gene have garnered attention due to their potential role in tissue remodeling and inflammatory responses [3].

In addition to MMP-8, other matrix metalloproteinases, such as MMP−1, MMP−2, MMP−9, and MMP−13, have also been implicated in tissue remodeling and peri-implant inflammation. These MMPs participate in the degradation of various extracellular matrix components and are modulated by complex interactions with cytokines and bacterial products [3].

The peri-implant sulcus in a healthy state comprises a thin, non-keratinized junctional epithelium with a minimal inflammatory cell infiltrate and perpendicular collagen fiber bundles, providing a barrier against microbial invasion [1]. In contrast, peri-implantitis is associated with the apical migration of the junctional epithelium, an extensive inflammatory infiltrate rich in neutrophils and macrophages, increased vascularization, and parallel-oriented collagen fibers that offer less resistance to bacterial penetration [1]. Microbiologically, diseased peri-implant sites harbor a dysbiotic biofilm that is distinct from healthy implants, characterized by a higher prevalence of anaerobic Gram-negative organisms, although with compositional differences compared to periodontitis [2]. These histological and microbial differences underscore the complex host response and provide the rationale for the exploration of genetic susceptibility factors such as MMP−8 polymorphisms.

MMP−8, also known as neutrophil collagenase, is a zinc-dependent protease that plays a crucial role in degrading type I collagen, a key structural component of periodontal and peri-implant tissue [4,5]. Elevated levels of MMP−8 in peri-implant sulcular fluid (PISF) during disease states, such as peri-implant mucositis and peri-implantitis, highlight its involvement in tissue degradation and inflammation [6,7]. Compared to its latent form, active MMP−8 (aMMP−8) is a more reliable biomarker of active disease processes, indicating ongoing collagen breakdown and inflammation [8,9].

Recent advancements, including aMMP-8 chairside diagnostic tests, have shown promising sensitivity and specificity in detecting early peri-implant disease [5,7]. While extensive research has established MMP-8 as a key mediator of connective tissue destruction and inflammation regulation in periodontitis, fewer studies have examined its role in peri-implantitis [10].

Previous studies have investigated single-nucleotide polymorphisms (SNPs) in the MMP−8 gene, such as −799C/T (rs11225395), −381A/G (rs1320632), and +17C/G (rs2155052), as potential genetic factors influencing susceptibility to periodontitis [11,12]. The −799C/T polymorphism in the promoter region is particularly noteworthy due to its association with increased transcriptional activity and heightened MMP-8 expression, with the T allele linked to a higher risk of inflammatory tissue destruction [12,13]. However, data on the role of these polymorphisms in peri-implantitis remains limited.

This study aimed to investigate the correlation between the distribution of MMP-8 gene variants−799C/T (rs11225395), −381A/G (rs1320632), and +17C/G (rs2155052)—and the clinical parameters of peri-implant disease.

## 2. Materials and Methods

### 2.1. Study Design and Ethical Approval

This cross-sectional study was conducted at the Department of Preventive Dentistry, Periodontology, and Implant Biology, School of Dentistry, Aristotle University of Thessaloniki, Greece. Participants were recruited based on predefined inclusion and exclusion criteria. The study was approved by the Ethical Committee of the School of Dentistry, Aristotle University of Thessaloniki, Greece (approval number: 115/25-05-21) and registered in the ClinicalTrials.gov database (https://clinicaltrials.gov) (ID: NCT05711407). All procedures adhered to the Declaration of Helsinki, and written informed consent was obtained from all participants.

### 2.2. Participant Selection

Participants were classified according to the 2018 Classification of Periodontal and Peri-Implant Diseases [1]. Peri-implant health was defined as the absence of clinical signs of inflammation and no radiographic bone loss. Peri-implant mucositis was defined as bleeding on probing without bone loss. At the same time, peri-implantitis included bleeding and/or suppuration on probing, an increased probing depth compared to previous measurements, and radiographic bone loss beyond initial remodeling [1].

The study included 120 participants, divided into two groups based on the 2018 classification criteria for periodontal and peri-implant diseases: those with peri-implantitis and those with healthy implants or peri-implant mucositis [1].

The inclusion criteria required participants to be over 30 years old, free from systemic conditions or medications affecting the periodontal tissue, and to have had no periodontal treatment or antibiotic use in the past six months. Exclusion criteria included systemic infections (e.g., hepatitis, HIV), pregnancy or lactation, and compromised DNA quality for laboratory analysis.

The initial sample size was determined based on previous research on the −799C/T (rs11225395) SNP in the *MMP-8* gene, which reported a minor allele frequency (MAF) of approximately 40–43% and an odds ratio (OR) of 1.5–2.0 for its association with peri-implantitis and periodontitis [13]. Using these parameters, a case–control design with an equal distribution of cases and controls required a minimum of 45 participants per group to achieve 80% power (1−-β = 0.8) at a significance level of 0.05. The calculation incorporated genotype frequencies from prior studies and targeted moderate effect sizes to identify significant differences in the peri-implantitis risk. The sample size was increased to accommodate potential variability and allow subgroup analyses to enhance the study’s robustness.

### 2.3. Clinical Assessment

The distance from the shoulder of the prosthetic crown to the mucosal margin was also measured. Additionally, the clinical attachment level (CAL) was defined as the distance from the shoulder of the prosthetic crown to the base of the sulcus or peri-implant pocket. All measurements were taken at six sites per implant using a 15 mm periodontal probe (Hu-Friedy^®^ CP-12, #30) with 1 mm increments. Measurements were repeated after the prosthesis was removed. All examinations were conducted by the same examiner (I.F.), and the intra-examiner reproducibility was assessed during two calibration sessions. The intra-examiner agreement, evaluated using the intraclass correlation coefficient (ICC), showed a high level of agreement (0.93; 95% CI: 0.89 to 0.96). Unstimulated saliva and oral rinse samples were collected in the morning after participants had refrained from eating, drinking, smoking, or brushing their teeth for at least one hour.

### 2.4. DNA Extraction and Genotyping

Saliva samples were collected and stored at −80 °C until DNA extraction was performed using the Quick-DNA Miniprep Plus Kit (Zymo Research, Irvine, CA, USA), following the manufacturer’s protocol. DNA quality was confirmed via spectrophotometric measurement at 310 nm.

### 2.5. Polymerase Chain Reaction (PCR) and Sequencing

The polymerase chain reaction (PCR) was performed in 25 µL reaction volumes, which included 2 µL of template DNA, specific primers, dNTPs, Taq polymerase, and buffer. The amplification protocol involved an initial denaturation step at 95 °C for 2 min, followed by 30 cycles of 95 °C for 30 s, 62 °C for 1 min, and 72 °C for 1 min, concluding with a final extension at 72 °C for 5 min. PCR products were confirmed by agarose gel electrophoresis and purified using the Nucleospin Extract II kit (Macherey-Nagel, Düren, Germany). After amplification, the DNA products were further purified using the DNA Clean & Concentrator-5 kit (Zymo Research, Irvine, CA, USA). This kit effectively removes salt, primers, enzymes, and other contaminants, allowing for the rapid purification and concentration of up to 5 µg of DNA from various enzymatic reactions.

The purified DNA was then sequenced at Eurofins Genomics Europe Pharma and Diagnostics Products & Services Sanger/PCR GmbH, an ISO/IEC 17025:2018 accredited laboratory. The sequencing was conducted using high-precision Sanger sequencing to analyze the −799C/T (rs11225395) and −381A/G (rs1320632) polymorphisms in the MMP-8 gene. The sequencing was performed on the PlateSeqSupreme platform, ensuring high accuracy and adherence to current scientific standards. The laboratory generated and validated the results.

The sequencing data were analyzed using the SnapGene^®^ software (version 7.0.0; Dotmatics, Boston, MA, USA) to visualize and interpret the results. SnapGene^®^ was used to verify the quality of the sequencing reads, align the sequences, and confirm the presence of specific polymorphisms.

### 2.6. Statistical Analysis

Statistical analyses assessed the relationships between SNP genotypes, allele-specific risks, the presence or absence of peri-implantitis, and clinical parameter evaluations. All analyses were performed using SPSS Statistics^®^ Version 29 (IBM Corp., Armonk, NY, USA). A chi-squared test for independence was used to determine whether the distribution of peri-implantitis cases differed significantly among the three SNP799 genotypes (C, C/T, and T). This test compared the observed and expected frequencies of peri-implantitis cases within each genotype group, with statistical significance set at *p* < 0.05.

The C and T alleles’ frequencies were calculated to assess specific risks and each allele’s presence stratified diagnosis rates. Logistic regression analysis was performed to evaluate the association between the presence of the T allele and the likelihood of peri-implantitis. In this analysis, the dependent variable was the diagnosis of peri-implantitis (binary: 0 = health, 1 = peri-implantitis), and the independent variable was the presence of the T allele (binary: 0 = absent, 1 = present). The statistical output included odds ratios (ORs) with 95% confidence intervals (CIs) and *p*-values.

Furthermore, clinical parameters such as the mean probing depth (PD), clinical attachment level (CAL), and bleeding on probing (BOP) were compared across the three SNP genotypes using the Kruskal–Wallis test, a non-parametric method suitable for continuous variables. Descriptive statistics for these parameters were reported as the mean ± standard deviation (SD). A *p*-value of <0.05 was considered statistically significant for all analyses. This approach comprehensively evaluated both genotype-specific and allele-specific effects on peri-implantitis and its associated clinical parameters. The study flowchart is depicted in Figure 1.

## 3. Results

Out of the initial 120 samples, 95 were selected for analysis. In comparison, the remaining 25 samples were excluded due to low-quality sequencing results—specifically, those with a contig read length (CRL) of 100 or less. At the -381A/G (rs1320632) locus, six samples (6.3%) were heterozygous (AG), while the remaining 89 samples (93.7%) were homozygous for the A allele. In contrast, at the -17C/G locus, all samples were homozygous for the C allele. As a result, the analysis focused on the −799C/T SNP, which exhibited variants within the study population. Of the 95 participants in the final analysis, 46 were classified as having peri-implantitis. 

### 3.1. Demographic Characteristics of the Participants

The distribution of the demographic parameters, including sex and smoking status, was similar across the SNP genotypes (genotype C, genotype C/T, and genotype T), with no statistically significant differences observed. Similarly, the clinical parameters, such as PD, CAL, and BOP, were compared across the three SNP genotypes (genotypes C, C/T, and T). While slight variations were noted, such as higher mean PD and CAL values in individuals with genotype T, no statistically significant differences were found for the clinical parameters. These results suggest that the SNP799 genotype may not significantly impact the recorded clinical parameters (Table 1 and Table 2).

### 3.2. Association of the −799C/T (rs11225395) Genotypes and Allele Frequencies with Peri-Implantitis and Health Diagnosis

The diagnosis rates for peri-implantitis were 33.3% for genotype C, 57.1% for genotype C/T, and 64.0% for genotype T. A chi-squared test revealed a statistically significant association between the genotype and the diagnosis of peri-implantitis (χ^2^ = 7.11, *p* = 0.0286) (Table 3). The frequency of the C allele was 56.9%, while the T allele accounted for 43.1% of all alleles. Individuals with the T allele had a peri-implantitis diagnosis rate of 60.4%, compared to 42.9% for those with the C allele. The logistic regression analysis confirmed the significant association between the T allele and peri-implantitis. The T allele was linked to substantially higher odds of developing peri-implantitis, with an odds ratio (OR) of 3.04 (*p* = 0.010). This finding highlights a strong genetic predisposition associated with the T allele. The analysis showed a coefficient of 1.114 for the T allele, with a standard error of 0.431, and the 95% confidence interval (CI) for the OR ranged from 1.31 to 7.10. The model’s constant term had a coefficient of -0.693, a standard error of 0.327, and a *p*-value of 0.034. These results underscore the robust relationship between the T allele and an increased susceptibility to peri-implantitis.

## 4. Discussion

This study examined the relationship between specific polymorphisms of the MMP-8 gene (−799C/T, −381A/G, and +17C/G) and peri-implantitis, as well as their correlation with clinical parameters such as the probing depth (PD), clinical attachment loss (CAL), and bleeding on probing (BOP). The results highlighted the −799C/T polymorphism, particularly the T allele, as a potential genetic factor in peri-implantitis. This allele has been previously implicated in periodontitis and peri-implant tissue destruction due to its upregulation of neutrophil-derived MMP-8, which is critical in degrading type I collagen and initiating inflammatory cascades. However, no associations were found for the −381A/G and +17C/G polymorphisms, as most subjects displayed homogenous SNP variants at these positions. These findings support and expand upon previous research on MMP-8 genes in periodontitis, emphasizing their critical role in tissue remodeling and inflammation. They also contribute to the limited studies on peri-implantitis [5,14].

Costa-Junior et al. (2012) investigated the effect of the −799C/T polymorphism on early implant failure, finding that individuals with the T allele were more susceptible to osseointegration failure [10]. The current study builds on these findings by showing that the T allele is associated with peri-implant inflammation, highlighting its relevance across various stages of implant failure [10]. These findings support a continuum model where genetic factors, such as the −799C/T polymorphism, predispose individuals to early osseointegration failure and long-term inflammatory complications, including peri-implantitis. This continuum emphasizes the importance of host susceptibility from the initial healing phase through the maintenance period.

Research on periodontitis highlights the significance of the −799C/T polymorphism in influencing the disease risk. For instance, Chou et al. (2011) and Emingil et al. (2014) identified strong associations between the T allele and periodontitis, linking this genetic variant to elevated MMP-8 levels and increased tissue damage [12,13]. The current findings suggest that the effects of this polymorphism may extend beyond the periodontal tissue, affecting peri-implant environments where inflammation and bone loss occur through similar, although not fully understood, pathogenetic mechanisms. However, the peri-implant environment lacks a periodontal ligament and shows reduced vascularization, which may modulate the expression and impact of inflammatory mediators, including MMP-8, despite the shared genetic predisposition. The common genetic risk factors between early implant failure and peri-implantitis suggest that the −799C/T polymorphism could have a broader impact across various stages of implant-related complications.

An analysis of the clinical parameters across genotypes revealed no statistically significant differences in the probing depth (PD), clinical attachment level (CAL), or bleeding on probing (BOP) between the healthy and diseased groups. Although the T allele group exhibited numerically higher mean values for all three parameters, these differences were insignificant (*p* > 0.05). This suggests that, while the −799C/T polymorphism may influence the susceptibility to peri-implantitis, its impact on the clinical severity in this cohort remains inconclusive. It is also plausible that the clinical expression of the −799T allele may be modulated by environmental factors such as plaque control, systemic health, or epigenetic regulation, which were not assessed in the current design.

These findings contrast previous research on periodontitis, which reported significant correlations between the −799C/T polymorphism and worse clinical parameters [11,12,15,16,17]. This discrepancy may be attributed to differences in the sample size, population characteristics, or the distinct inflammatory environments of periodontitis and peri-implantitis.

Future studies with larger sample sizes and more diverse populations must further clarify the relationship between the −799C/T genotype and clinical outcomes in peri-implantitis. This study also emphasizes the limited research on MMP−8 genetic variants in peri-implantitis compared to periodontitis, highlighting the need for further investigation into the genetic and molecular pathways specific to peri-implantitis, particularly concerning implant-specific factors. Future risk models may also incorporate polygenic risk scoring (PRS), which can account for multiple SNPs to generate a more robust individual risk profile.

Identifying the −799C/T polymorphism as a genetic risk factor for peri-implantitis has important clinical implications. Genetic screening for this polymorphism could help to identify individuals at higher risk for peri-implant disease, enabling targeted preventive strategies such as more frequent maintenance visits and early interventions. Moreover, the strong correlation between aMMP−8 levels and clinical parameters supports the use of aMMP−8-based diagnostic tools for the early detection and monitoring of peri-implant disease progression. These chairside tests have shown high sensitivity and specificity in detecting active disease, offering a practical solution to improve patient outcomes [7,18].

Although this study focused specifically on the −799C/T polymorphism of the MMP−8 gene, it must be acknowledged that peri-implantitis is a multifactorial disease influenced by a broader network of host factors. Other members of the matrix metalloproteinase family, such as MMP−1, MMP−9, and MMP−13, have been implicated in connective tissue breakdown and bone resorption in both periodontitis and peri-implantitis [14]. Their roles, along with the regulatory influence of tissue inhibitors of metalloproteinases (TIMPs), suggest a complex proteolytic balance that modulates tissue degradation [13]. Additionally, cytokine gene polymorphisms—especially in IL−1β, TNF−α, and IL−6—as well as variants in toll-like receptors (e.g., TLR2, TLR4) and components of the RANK/RANKL/OPG pathway, have shown associations with peri-implant disease susceptibility and severity [2,14]. These molecules collectively influence the host inflammatory response and bone homeostasis. Comparing the MMP−8 polymorphism to these established factors highlights its potential as a diagnostic marker but also underlines the need for integrated multi-marker approaches in future risk assessment models. Such approaches could enhance the predictive accuracy for peri-implantitis and inform precision preventive strategies tailored to individual genetic and inflammatory profiles.

This study has several limitations. The small sample size may have limited the statistical power to detect associations, especially for the −381A/G and +17C/G polymorphisms. The study population’s homogeneous nature also limits the findings’ generalizability to other ethnic groups. Another limitation is the lack of a comparative analysis with other MMPs or pro-inflammatory mediators, which could provide a more comprehensive view of the multifactorial host response. Future studies should incorporate multi-marker panels or polygenic risk scores to assess individual susceptibility better. Furthermore, the cross-sectional design prevents causal inferences about the relationship between genetic polymorphisms and peri-implantitis. Future studies should address these limitations by including larger, more diverse cohorts and employing longitudinal designs.

## 5. Conclusions

This study highlights the possibly critical role of the −799C/T polymorphism in MMP−8 as a genetic determinant of peri-implantitis, extending findings from periodontitis and early implant failure to chronic peri-implant disease. These findings emphasize the need for more targeted research on MMP−8 in peri-implant diseases and support the integration of genetic screening and biomarker-based diagnostics into personalized implant care.

## Figures and Tables

**Figure 1 jpm-15-00182-f001:**
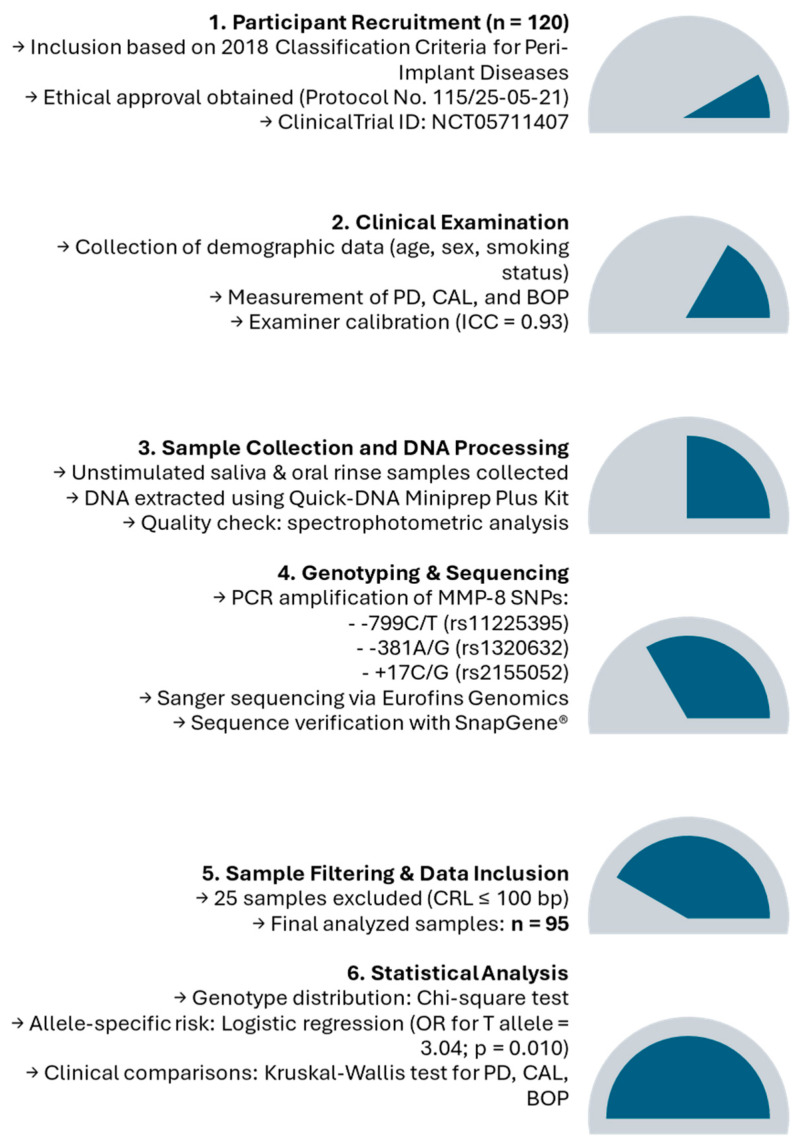
Research framework of the study on the MMP-8 -799C/T polymorphism and peri-implantitis.

**Table 1 jpm-15-00182-t001:** Demographic characteristics of the participants.

Demographic Parameter	Genotype C (*n* = 42)	Genotype C/T (*n* = 28)	Genotype T (*n* = 25)	Total (*n* = 95)	*p*-Value
Sex, n (%)					0.981 ^a^
Male	23 (54.8%)	18 (64.3%)	15 (60.0%)	56 (58.9%)	
Female	19 (45.2%)	10 (35.7%)	10 (40.0%)	39 (41.1%)	
Smoking status, n (%)					0.252 ^a^
Non-smokers	24 (57.1%)	16 (57.1%)	15 (60.0%)	55 (57.9%)	
Smokers	18 (42.9%)	12 (42.9%)	10 (40.0%)	40 (42.1%)	
Age (years, Mean ± SD)	58.3 ± 10.21	61.2 ± 9.76	62.5 ± 8.45	-	0.734 ^b^

^a^ Chi-squared test. ^b^ Mann–Whitney U test.

**Table 2 jpm-15-00182-t002:** Comparison of clinical parameters across SNP genotypes (genotype C, C/T, T).

Clinical Parameter	Genotype C (*n* = 42)	Genotype C/T (*n* = 28)	Genotype T (*n* = 25)	Total (*n* = 95)	*p*-Value
Mean Probing Depth (PD, mm)	3.49 ± 1.10	3.56 ± 1.14	4.09 ± 1.45	3.71 ± 1.23	0.335
Mean Clinical Attachment Level (CAL, mm)	3.76 ± 1.15	3.78 ± 1.25	4.61 ± 2.26	4.02 ± 1.56	0.531
Bleeding on Probing (BOP, %)	46.04 ± 25.79	36.12 ± 33.93	48.82 ± 36.77	43.0 ± 32.16	0.103

Statistical significance was assessed using the Kruskal–Wallis test, with *p*-values reported.

**Table 3 jpm-15-00182-t003:** Genotype frequencies and diagnosis outcomes (peri-implantitis vs. health).

Genotype	Frequency (*n*)	Peri-Implantitis, *n* (%)	Health, *n* (%)
CC	42	14 (33.3%)	28 (66.7%)
C/T	28	16 (57.1%)	12 (42.9%)
TT	25	16 (64.0%)	9 (36.0%)

## Data Availability

The data supporting this study’s findings are openly available.

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
