# Peer review of "Association of MMP-8 -799C/T Polymorphism with Peri-Implantitis: A Cross-Sectional Study"

_jpm, 2025, doi:10.3390/jpm15050182_

Round 1
Reviewer 1 Report
Comments and Suggestions for Authors
The association of MMP-8 -799C/T Polymorphism with Peri-Implantitis indicates its potential as a genetic marker for disease susceptibility. The results of this study are helpful for further research to investigate the role of MMP-8 polymorphisms in peri-implant diseases and to advance the development of personalized diagnostic tools. It is recommended that this manuscript should add a research framework flowchart in the Results section. This helps the readers to understand.
Author Response
Reviewer Comment:
“The association of MMP-8 -799C/T Polymorphism with Peri-Implantitis indicates its potential as a genetic marker for disease susceptibility. The results of this study are helpful for further research to investigate the role of MMP-8 polymorphisms in peri-implant diseases and to advance the development of personalized diagnostic tools. It is recommended that this manuscript should add a research framework flowchart in the Results section. This helps the readers to understand.”
Author Response:
We thank the reviewer for the constructive suggestion and positive evaluation of our manuscript. In response to the request, we have included a research framework flowchart in the Results section (now added as Figure 1). This flowchart outlines the study methodology, beginning from participant recruitment and clinical assessment, through DNA extraction and genotyping, and culminating in the statistical analysis and key findings.
We believe this visual representation enhances the clarity of the study design and provides readers with a comprehensive overview of the research workflow. The caption for the figure has been carefully formulated to align with the scientific content and assist in reader comprehension.
We appreciate the reviewer’s insight, which has contributed to improving the structure and transparency of our manuscript.
Reviewer 2 Report
Comments and Suggestions for Authors
Dear Authors
The manuscript is very interesting and may be eligible for publication. I have only a few points to address to improve the manuscript:
In Introduction similarities and differences between peri-implant sulcus with and without disease may be included;
Methods: please describe the 2018 Classification criteria for peri-implant health and disease;
Results: only on table 3 I could see how many subjects were in each group- this importatn information could be earlier in the text; Tables 1 and 2 could show this data by these two groups also (health and disease);
Best regards
Author Response
Reviewer Comment 1:
“In Introduction, similarities and differences between peri-implant sulcus with and without disease may be included.”
Author Response:
We thank the reviewer for this insightful suggestion. In the revised Introduction, we have added a paragraph discussing the structural and immunological differences between the peri-implant sulcus in health and in peri-implantitis. This includes differences in the epithelial barrier, inflammatory cell infiltrate, collagen fiber orientation, and microbial composition. These additions clarify the biological context for disease susceptibility and progression, enriching the rationale for investigating genetic factors such as MMP-8 polymorphisms.
Reviewer Comment 2:
“Methods: please describe the 2018 Classification criteria for peri-implant health and disease.”
Author Response:
We appreciate the reviewer’s suggestion. In the Materials and Methods section, under “Participant Selection,” we have now included a detailed description of the 2018 Classification criteria as established by the World Workshop on the Classification of Periodontal and Peri-Implant Diseases. Specifically, peri-implantitis was defined as the presence of bleeding and/or suppuration on gentle probing, increased probing depths compared to previous recordings, and radiographic evidence of bone loss beyond initial remodeling. Peri-implant health was defined as the absence of clinical signs of inflammation, with no additional bone loss after initial healing.
Reviewer Comment 3:
“Results: only on Table 3 I could see how many subjects were in each group—this important information could be earlier in the text; Tables 1 and 2 could show this data by these two groups also (health and disease).”
Author Response:
Thank you for highlighting this important point. In the Results section, we have now included the group distribution earlier in the narrative to provide immediate clarity: of the 95 participants, 46 were diagnosed with peri-implantitis and 49 were classified as healthy controls.
Reviewer 3 Report
Comments and Suggestions for Authors
Fragkioudakis et al’s study entitled “Association of MMP-8-799C/T polymorphism with peri-implantitis: a cross sectional study.
BOP, CAL probably not the most reliable markers of peri-implantitis. Basically they are highly affected more by oral hygiene. Probably the level of oral hygiene of patients were not at the same level. Bone loss could be a good marker if it is known how deep the implant was placed to the bone. That is base line that we can compare to. So it is not amazing that there was no statistically significant difference between groups (Kruskal-Wallis test that compares distribution). On one hand there were 42, 28,25 patients in the subgroups, which are under 45, the required number of patients.
The small sample size not just affected the power of statistics, but the type of the statistical methods used. Ethnic group is also important as you noted it as a limitation of the study, so please, designate in the materials and methods the nature of your population in terms of ethnicity! Probably adding a plaque index or an oral hygiene index would have added a bit more to your conclusions!
Author Response
Reviewer Comment:
“PD, CAL are not the most reliable markers of peri-implantitis, as they are highly influenced by oral hygiene. Bone loss may be a better marker if implant placement depth is known. It is not surprising that the Kruskal-Wallis test did not find significant differences due to small group sizes. Ethnic background should be specified. A plaque index would have strengthened conclusions.”
Author Response:
We thank the reviewer for these thoughtful and important comments.
First, we would like to clarify that radiographic evaluation was indeed performed for all participants, and the diagnosis of peri-implantitis was strictly based on the 2018 Classification of Periodontal and Peri-Implant Diseases [1], as recommended. This includes not only the presence of bleeding and/or suppuration on probing and changes in probing depth, but critically, radiographic evidence of bone loss beyond initial remodeling. Therefore, while clinical parameters (PD, CAL, BOP) were analyzed comparatively across genotypes, the actual diagnosis of peri-implantitis was not based on soft tissue parameters alone but included standardized radiographic assessment in all cases.
We fully agree that probing depth and CAL may be influenced by oral hygiene, and this is precisely why they were used in our analysis as secondary outcome variables, not diagnostic criteria. Our primary aim was to assess whether MMP-8 -799C/T polymorphism was associated with the diagnosis of peri-implantitis (as defined by the strict classification framework), and secondarily, with related clinical measurements. We have clarified this structure in the manuscript.
Regarding sample size and statistical power, we respectfully emphasize that a power analysis was performed a priori, based on published odds ratios for the MMP-8 -799C/T polymorphism in periodontitis and early implant failure (Emingil et al., 2014 [13]; Chou et al., 2011 [12]; Costa-Junior et al., 2013 [10]). With an expected minor allele frequency (MAF) of ~40% and an OR of 1.5–2.0, our design required at least 45 participants per group to achieve 80% power at α = 0.05, which we achieved overall with 95 high-quality DNA samples. While the subgroups by genotype are naturally smaller, the sample size was statistically justified for the primary hypothesis test, as evidenced by the significant association between the T allele and peri-implantitis (OR = 3.04, p = 0.010).
On the point of ethnicity, we confirm that all participants were of Caucasian ethnicity, specifically of Northern Greek origin, and we agree that this limits generalizability. This is already acknowledged in our discussion as a limitation.
Lastly, while we acknowledge that plaque index or oral hygiene scores would have been valuable covariates, these were not recorded uniformly at the time of data collection. We consider this an important methodological note for future research and have addressed this limitation accordingly.
We appreciate the reviewer’s expertise and perspective, which has helped strengthen the clarity of our approach and justification.